# Experience of Good Practice in an Occupational Accident Mutual Insurance Society, Based on the Voice of Patients and Professionals

**DOI:** 10.3390/ijerph16203856

**Published:** 2019-10-12

**Authors:** Diego Moya, Mar Iglesias, Rafael Manzanera, Fernanda de la Torre, Manel Plana, Gloria Gálvez, Mercedes Guilabert

**Affiliations:** 1Health and Economic Benefits Department, Mc-Mutual. Barcelona, 08037 Cataluña, Spain; dmoya@mc-mutual.com (D.M.); miglesias@mc-mutual.com (M.I.); rmanzanera@mc-mutual.com (R.M.); fdelatorre@mc-mutual.com (F.d.l.T.); mplana@mc-mutual.com (M.P.); 2Hospital Vall Hebron, Barcelona, 08035 Cataluña, Spain; ggalvez@vhebron.net; 3Health Psychology Department, Universidad Miguel Hernández, Elche, 03202 Comunidad Valenciana, Spain

**Keywords:** patient-centered care, patient outcome assessment, patient satisfaction, mutual insurance societies

## Abstract

In Spain, the protection of workers’ health is organized through what are known as occupational accident mutual insurance societies. While health protection is a fundamental issue within a mutual society, dimensions, such as patient quality and safety, are measured in the same way as in the conventional healthcare sector. However, in mutual societies, it is traditionally acknowledged that experiences of medical evaluation systems of healthcare provision and quality improvement are less frequent. The following is an example of how a Quality Plan has been structured within an organization with these characteristics, and instruments and measures have been developed to capture information in decision making from the perspective of patients and professionals. The Quality Plan represents the ongoing commitment of this organization to achieve patient-centered care. These changes revolve around these measures and, therefore, it is defined as a good practice.

The following manuscript has been drafted according to the recommendations of “A new structure for quality improvement reports” [1].

## 1. Brief Description of Context

Accidents at work and occupational diseases represent a worrying reality throughout the world. According to the ILO (International Labor Organization), in its Centenary Report [2], there were more than 374 million occupational accidents worldwide. These accidents affected around 10% of workers, of which 50% were mild, 25% were moderate, and 25% were serious, with 7500 daily deaths associated with accidents and occupational diseases, representing a rate of 11 per 100,000 workers, ranging from 3 in Europe to 17 in Africa. The estimated loss in workdays related to occupational health and safety accounts for about 4% of world GDP, a figure that can reach 6% in some countries [2].

According to Eurostat [3], 10 % of the working population covered by Mutual Societies partnering with the Spanish National Health System (MCSS) is treated annually due to an accident at work or an occupational disease. In Europe as a whole, the frequency of accidents in which the worker required more than four days of leave ranged between 519 per 100,000 workers in Greece and 874 in Sweden. The protection system is different in each country, but the purpose of the mutual benefit system shares the same objectives [4]. 

The protection of the worker against situations of occupational accident or disease, appeared as a general directive in Europe at the beginning of the 1970s [5], although it is the Single European Act [6] that explicitly obliged states to establish provisions to protect the health of workers and the improvement of the work environment. In the late 1980s, directives appeared in order to harmonize minimum policies in the laws of the member states. Corporate responsibility is an essential element in workers' protection regulations. Labor law in different countries regulates, with some peculiarities, the characteristics and limitations of corporate responsibility. There are also various ways in which healthcare and economic coverage of labor contingencies are financed and provided. The concept of work injury is somewhat different according to the countries considered [7] both in terms of the conditions required and in situations excluded from the concept. This heterogeneity makes international comparisons very difficult. When trying to carry out a comparative examination of the protection of occupational accidents and diseases in the countries of the European Union, there was a greater convergence in the coverage and financing systems of economic benefits [8]. Except in Holland, where there is no specific coverage, in other countries, there is. In most European countries, specific funds are allocated from companies that are managed through insurance companies, offering an ample and more extensive health care than the general one, without waiting periods, with powerful devices to facilitate return to work and lists of occupational diseases. 

Many of these features are given in Spain, which, nonetheless, maintains private non-profit organizations and business groups, which specifically manage the quotas that employers and workers contribute to the Social Security and that have specialized health resources in traumatology and rehabilitation, both on an outpatient and hospital basis. In Spain, these organizations are called “Mutuas” (mutual societies).

In Spain, current mutual societies manage around ten billion euros covering expenses related to the coverage of accidents at work, occupational diseases, and the control of benefits for temporary disability due to non-work-related illness, among other benefits, which represents 1% with respect to the GDP (Gross Domestic product). To have a frame of reference, we can say that, in relation to health expenditure, it represents 15% of public health expenditure and almost half of the volume of private health expenditure. The economic and reinsurance benefits represent the main expenditure of mutual societies, while health services account for less than 10% of the total expenditure [9,10,11].

Mutual societies take responsibility for the healthcare provision and the economic and partially preventive activity of the workers of companies affiliated to the mutual society, as well as self-employed workers. Specifically, within their portfolio of services are: temporary work disabilities (sick leave) related to accidents at work and occupational diseases (professional contingencies) that entail health care and the compensatory economic benefit arising from the inability to work; temporary work disability (sick leave) related to a non-work-related cause (common contingencies) that are monitored by the mutual society to control the economic benefit and health care provided in certain situations to expedite the return to work. Mutual companies also face maternity benefits at risk, help of parents of seriously ill hospitalized children and for the cessation of activity of self-employed workers. Common contingencies occur as a complex result of factors related to the illness, productive capacity, type of work, values of the person, company styles, and the problem-solving capacity of the public health system, among others. This complexity is also influenced by the conflict of interests between the perception of the worker and the companies. Women and young people tend to have more sick leave, but of shorter duration [12].

To this end, health professionals of mutual societies develop two types of key actions, on the one hand, health care, and on the other, evaluative action.

### 1.1. Healthcare

A set of actions related to health care provision granted to affected workers. This ranges from medical urgency to functional rehabilitation. It provides all the necessary means: medication, prostheses, diagnostic tests, surgical interventions, and those defined by law, depending on the type of contingency.

### 1.2. Evaluative Action

A set of evaluative actions to assess the ability to return to work that are required in conjunction with usual health care procedures. It relates, in an essential way, the injury or pathology with the conditions or requirements of the job, determining the situation of incapacity or not for a given type of work at all times.

In the data available for 2012 [4], Spanish mutual societies attended to 1,150,000 patients of a working population of about 12 million people. At the end of 2018, the size of the working population in Spain was 19 million workers. The typology of these accidents was diverse. Annually, 10% of people suffer a work accident. In 2013, 3.5% corresponded to accidents in the workplace, resulting in absence from work, 0.5% to accidents traveling to or from work requiring sick leave, and 6% to accidents not requiring sick leave. Mortality is less than 1 per 10,000. In terms of activity, more than 5 visits per 100 protected workers, 5 visits per injured worker, 5 surgical interventions per 1000 protected workers, and 70 interventions per 1000 injured workers were accounted for. 

The mutual society sector, since its inception in the early twentieth century, has undergone numerous changes in its legal framework, public control, territorial coverage requirements, dimensions, cooperation, and merger strategies. Company holders, representative trade unions, and public administrations involved have marked trends that do not always coincide but are aimed at achieving greater transparency and reinforcing public control and a greater tendency towards concentration.

## 2. Outline of the Problem

The working population represents a large part of Spain's population (19 million people employed in Spain in 2018) [13]. As already mentioned, in Spain, the protection of workers' health is organized through what are known as occupational accident mutual insurance societies. While health protection is a fundamental issue within a mutual society, dimensions such as patient quality and safety, are measured in the same way as in the conventional healthcare sector. However, in mutual societies, it is traditionally acknowledged that experiences of medical evaluation systems of healthcare provision and quality improvement are less frequent.

The following is an example of how a Quality Plan has been structured within an organization with these characteristics, and instruments and measures have been developed to capture information in decision making from the perspective of patients and professionals. The Quality Plan represents the ongoing commitment of this organization to achieve patient-centered care. These changes revolve around these measures and, therefore, it is defined as a good practice.

Stakeholders of Midat Cyclops MUTUAL (MC MUTUAL) society that are part of this organization and, therefore, the Quality Plan are described in the following Table 1 [14].

### Why Is a Quality Plan Made in a Mutual Society?

The Quality Plan of MC MUTUAL was born in 2013 (Health Quality Plan 2014–2016) from the vocation of its General Director to improve the health performance of a mutual society. No conflict had previously occurred, there had been no type of health care crisis; it was simply decided to take a step in the line of improving the health profile of a mixed organization, such as mutual societies, which have a dual orientation in both healthcare and economic aspects.

The Quality Plan has focused on achieving the satisfaction of two main key players: patients and professionals. The thematic actions have been oriented to five blocks of action:
Patient safety: actions aimed at promoting the use of the adverse event reporting system, consolidating risk management tools, and implementing the pending patient safety programs, evaluating their effectiveness. Right clinical practice: actions to reduce the variability in the most relevant clinical practices of MC MUTUAL (healthcare and evaluative), to optimize the adequacy of clinical practice based on evidence and to measure the opinion and experience of the patient.Knowledge management, actions aimed at defining the health care knowledge map, and promoting the dissemination of internal knowledge.Health innovation: actions aimed at implementing the model for evaluating health quality standards, promoting research, teaching, and partnerships with universities, and using digital transformation to improve health care and the implementation of the new Electronic Medical Record for improving the quality of healthcare.Change management: actions to develop certifications and accreditations in primary healthcare centers and hospital centers themselves, through integrated models of health quality, to promote the culture of healthcare quality and assess the impact on culture and to promote the involvement of healthcare providers in Healthcare Quality and establish new partnerships with providers and Mutual Societies that collaborate with the National Health System. 

Patient safety, right clinical practice, and knowledge management lines as blocks of action of the 2014–2016 and 2017–2019 Health Quality Plan, as a response to the traditional objectives of other health care organizations. Health innovation incorporated some healthcare innovations (telemedicine, tele-physiotherapy, 3D, and others), as well as those of a methodological and organizational nature (creation of specific standards for mutual societies, digitalization, patient experience, culture, and quality assurance, among others). Change management was considered as the line dedicated to converting those defined in the previous lines (creation of Health Quality References in health centers, certification, and accreditation, among others) into action and real practice. 

Patients and professionals are considered the two key players of the Health Quality Plan. Opinion surveys and patient experience studies, as well as quality assurance and culture studies, aimed at the opinion of professionals and were launched at the beginning of the Plan. Essential measures of the evolution of the Health Quality Plan were taken into consideration.

## 3. Key Measures for Improvement

Among the measures adopted to produce improvements within this organization and in accordance with its stakeholders, the following instruments stand out as periodic monitoring systems of the Mutual Society’s Health Quality Plan:

In the case of professionals: Quality Assurance and Safety Culture (Table 2).

### 3.1. Quality Assurance

The Quality Assurance questionnaire currently includes 24 questions (it initially had 21 questions), that explore eight dimensions, was also requested: Patient, Continuity, Adequacy, Technical Competence, Satisfaction, Accessibility, Clinical Safety, and Equity. This instrument monitored three groups of professionals. The first group was a set of 87 Health Quality References in the health centers. These references were designated by the entity in each of its centers where health care and evaluation activities were performed. The second group was a random selection of 54 professionals working in smaller centers (between four and seven professionals). This selection was based on the fact that it usually takes longer for information to reach smaller and peripheral centers. And the third group, a random selection of 100 professionals from the largest centers.

### 3.2. Safety Culture

The safety culture questionnaire initially included 10 items grouped into two dimensions [15]: five items that reflected favorable attitudes towards the safety culture in the organization and five items of a more instrumental nature.

Culture was addressed to all health professionals of the mutual society (800 professionals), including professionals from primary healthcare centers, hospital centers, and central services (managers).

In the case of patients, surveys of perception of the different services of a mutual society and patient experience were addressed.

### 3.3. Patient Perception Surveys

In the case of measures to incorporate the patient’s opinion, all perception surveys of the different services provided by a mutual society were included in order to incorporate the patient's voice in the evaluation of results. Qualitative studies of patient experience analysis were also included.

Satisfaction surveys are tools for measuring the opinion of patients that MC MUTUAL has used for years. Patient opinion studies in the organization began more than 15 years ago. 

The first surveys focused on measuring patient satisfaction when attending for work-related accidents, where the patient was physically administered the survey on discharge. The aspects that were evaluated were, information received, service, waiting time, and facilities. Currently, in primary health centers, the questionnaires have been addressed to patients undergoing medical supervision of their disability (CC-common contingency-), patients treated for accidents at work (PC-professional contingency-), patients in rehabilitation (RHB), and patients in the emergency department (EMG). In these surveys, traditional aspects, such as information received and service, have been maintained in the evaluation, and information on flexibility and adaptation, the comfort of the facilities, accessibility and proximity of the centers, and promptness/waiting times in the services have been included.

In hospital centers, the surveys were aimed at patients discharged with admission to the hospital centers during the period studied. The services that were evaluated by patients were hospitalization, surgery, rehabilitation, outpatients, and psychology (Table 2).

### 3.4. Patient Experience

Regarding the project to evaluate patient experience, focus groups were conducted on patients in the mutual society hospital centers in Barcelona, Madrid, and Seville in order to trace the positive and negative elements of the patients' experience during their health care process. The process focused on professional contingency, analyzing the experience of the patient from the occurrence of the accident in the workplace either in the company itself or traveling to or from work until the patient had completed the health care process either through discharge or permanent disability (Table 2).

## 4. Process of Gathering Information

In 2014, under the direction of a group of experts, the methodological bases were designed, and the studies of Quality Assurance and Safety Culture were adapted to the reality of the mutual society.

### 4.1. Quality Assurance

This study explored various specific aspects of the actions that the Health Quality Plan intended to address and that were related to the various action blocks indicated by the Health Quality Plan with the eight dimensions of health care quality with a global assessment of them. Note that in this instrument, the traditional 5-point response scale also includes the open valuations carried out by professionals (semi-structured surveys).

Five waves of the Quality Assurance survey were performed. The first four waves referred to the evolution of the 2014–2016 Health Quality Plan. In the fifth wave, the questionnaire was revised and adapted to the measurement needs of the new 2017–2019 Health Quality Plan; in this case, the questionnaire was extended to 24 items.

### 4.2. Safety Culture

The culture study has evolved from a 10-item questionnaire to a 12-item questionnaire, including four items of global aspects and patient safety, three of the instruments in my center, and four characteristics of the treatments applied. Finally, a global assessment of the quality level of the health care provided was proposed. Also, in this case, a traditional valuation of the options offered by the survey was encouraged, but in particular, the open valuations made by professionals (semi-structured surveys).

In the case of culture, four waves were performed. In the fourth wave, minor changes were incorporated into the questionnaire, taking advantage of the implementation of the new 2017–2019 Health Quality Plan, and it evolved into a questionnaire with 12 items. 

### 4.3. Patient Perception Surveys

In 2014, a review of satisfaction expectations of mutual society patients was carried out and the content, format, and time at which the questionnaires were administered (on patient discharge) based on focus groups, was also reviewed. From this review, updated content emerged in 2017 that was launched both in health centers and hospital centers.

The number of items has evolved as follows: the professional contingency questionnaire went from 7 to 27 items; the common contingency questionnaire went from 6 to 25 items; the rehabilitation questionnaire from 7 to 23 items, and the emergency questionnaire from 7 to 19 items.

### 4.4. Patient Experience

Studies focusing on analyzing patient experience in the mutual society began in 2017, starting off with health care centers and hospital centers at three important points nationwide. The content analysis of these experiences yielded a set of common elements requiring improvement; “never events” and “always events” refers to a practice or set of behaviors so relevant to patient experience that they should be guaranteed at all times, since they provide the basis for the relationship between patients and professionals and represent the organization’s ongoing commitment to patient-centered care [16].

## 5. Analysis and Interpretation

Regarding the studies of Quality Assurance and Safety Culture, studies were systematically carried out to analyze the data as well as the robustness of the instruments as measures to monitor quality and safety in the organization [15,17]. In 2018, a study was published [18] where the actions implemented in Quality Assurance and Safety Culture were evaluated.

In the case of patient perception surveys, around 3000 questionnaires have been used globally, targeting the five lines of patients commented on in health care centers and 1500 questionnaires in hospital centers. The survey evolved towards a high number of items (27 in the case of professional contingencies), which forced a discriminant validity study (as well as the detailed path model) among all the variables to be recognized in 2019, which really brought something different with regards to satisfaction and recommendation.

For guidance, Table 3 is attached, which indicates the overall satisfaction for each of the patient lines in the years 2014–2017.

Based on the results of the instruments used, specific indicators were used to assess the thematic areas of the Health Quality Plan, including the indicators of patient and professional perception as a result of the aspects assessed in the perception surveys of both groups. Some of the indicators are described below (Table 4):

## 6. Strategy for Change

Achieving patient-centered care is the purpose of improvement actions derived from the results of opinion polls, analysis of patient experience, and studies of Safety Culture and Quality Assurance. The work strategy initiated in 2014 has allowed the evolution of the results in this organization to be monitored.

The patients interviewed identified attributes that should meet the services they received, such as kindness, professionalism, and formality. The least valued attributes were flexibility, experience, proximity, and commitment. Innovation and proactivity were elements of improvement, according to the patients' perception. Regarding differences between health care provision, emergencies were highlighted as the service backed by most experience, work-related accidents as the service with greater professionalism, common contingencies as that offering the most commitment, and in the case of rehabilitation, proximity was highlighted.

The results have been exploited (systematically), analyzed (promoting actions), made public (in the corporate intranet and in its board committees), and in some cases, published in scientific journals.

The review of the questionnaires used in the opinion of the patients is underway, which will allow the use of new instruments (more specific and aimed at measuring whether patient expectations are met) and a new sampling strategy focused on patient profiles as opposed to the services used.

Experience studies have shown effects of significant strategic change; they have served locally (mobilization of the local organization for them to be carried out), regionally (regional managers value them as a competitive advantage for the organization), and as a company (for the cultural change that they represent and a certain reorientation of investment).

## 7. Effects of Change

According to the perception of the professionals in Quality Assurance, it was felt there had been a great improvement in evidence-based strategy and practice. There were also improvements, although to a lesser extent, in patient-centered care, waiting times, and equipment.

Health professionals on Culture considered there had been a marked improvement in strategy and, secondly, in patient-centered care.

An independent expert team defined the strength of the implemented actions (scope and intensity), and the congruence between the observed results and the implemented actions were analyzed. Great congruence (actions and results) was observed in strategy, clinical decision support systems, patient-centered care, and cost-effective treatments. They observed some congruence in equipment, monitoring, and waiting times. There was a lack of congruence and, therefore, an important target of action in evidence-based practice.

The use of data referring to patient satisfaction surveys has been carried out with less intensity. Since 2018, they have been systematically reviewed and some measures were taken by central management of the hospital centers and to a lesser degree by the central management of health care centers.

The patient experience sessions carried out involved subsequent work of technical reviews of patient´s opinions, with the adaptation of their comments to the organizational reality and the proposal of action measures, which were largely developed in the centers and cities where the patient experience sessions took place.

In the attached scheme (Figure 1), and as an example of impact, the projects and actions implemented and the measurement of their impact are shown. At the top, there is the UNE 179003 certification of risk management for patient safety [19] (from May 2015 to July 2016), which covered the 87 primary healthcare centers throughout Spain and the two hospital centers in Barcelona. The major actions that were launched at that time were the creation of the health quality references, the review of the corporate directly linked to clinical actions (Adverse Event Reporting System (AERS), hand hygiene, patient identification, falls, among others). The training program on the quality plan, patient safety, and the adverse event reporting system, aimed at the entire organization, was also launched. Some specific actions to highlight were the start of participation in the Quality Congresses of the Spanish Society of Healthcare Quality (SECA) and the implementation of the project to welcome new professionals, along with those discussed above. Studies on culture were carried out at the beginning and end of the period comprising April 2015–November 2016. In the same period, three waves of the Quality Assurance study were carried out: May 2015, November 2015, and May 2016. With the help of these studies, aspects of health quality were identified in which improvements were detected, some more immediate and others more in the medium term. In summary, those actions in which some impact was detected (Risk Map, Adverse Event Reporting System, Training, and Induction Plan) and in which it was not detected (Communication and Teaching/Research) during the period studied are highlighted at the end of the figure. 

## 8. Next Steps

### 8.1. What Has Changed with These Measures?

There has been a consolidation of the Safety Culture and Quality Assurance; in the case of culture, the study is systematically carried out every two years and the items are reviewed periodically. In the case of Quality Assurance, the study is carried out annually and the items are always reviewed in accordance with the current Quality Plan. 

On the other hand, patient experience and opinion measurement systems have been consolidated. As for the patient experience studies, once the first pilots have been carried out, studies are intended to be carried out in a decentralized manner, in primary healthcare centers, and hospital centers by properly trained health personnel. The program of patient opinion surveys is under review: format, sampling, data exploitation, analysis, and proposals for actions.

### 8.2. Lessons Learned

It has been found that a mutual society can be an organization capable of leading quality healthcare projects, which means an active commitment to doing, doing it well, and sharing it. Hence, the importance of being able to disseminate all actions.

Patients should be the pivot of this organization and hence, the importance of being able to share the maximum number of decisions with them regarding their health.

All these measurement systems are compatible and synergistic with all the dimensions of quality, especially those related to efficiency.

All the measures used in these studies have their limitations; they are measures based on the perception of the people who form part of this organization and of the people who receive the service. Despite this limitation, an attempt has been made to implement actions and see if they are perceived.

Finally, an integration of the measuring instruments of both patients and professionals is planned. The traceability between Safety Culture and Quality Assurance, the opinion surveys of patients, and the narrative of their experience are being reviewed, in short, to be able to establish links between the perception of the patients and the professionals that are part of this organization.

## Figures and Tables

**Figure 1 ijerph-16-03856-f001:**
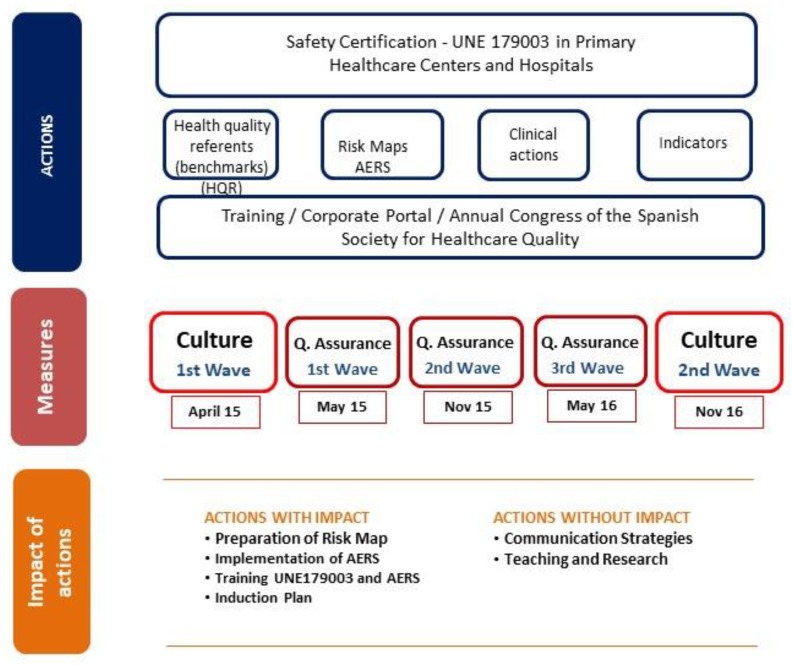
Actions implemented and the measure of their impact.

**Table 1 ijerph-16-03856-t001:** MC-MUTUAL Stakeholders.

Stakeholder	Category	Issues Considered Priority by Interest Groups
Human team of MC MUTUAL and its representatives	Professionals	The promotion of the effectiveness and efficiency of the social protection system. The contribution to the development of corporate social responsibility. The establishment of good communication with all the groups with which it interacts The promotion of equal opportunities for all. Transparent management system.
Governing bodies		
Sector of mutual insurance societies partnering with the Spanish NHS		
Companies and self-employed partners	Patients	
Users and beneficiaries and protected population		

**Table 2 ijerph-16-03856-t002:** Summary of studies conducted, instruments used, and target population.

Safety Culture			Quality Assurance		
Stakeholder	Dimensions	Items	Stakeholder	Dimensions	Items
All health professionals of the mutual society (800 professionals) including professionals from primary healthcare centers, hospital centers and central services (managers).	5 items that reflected favorable attitudes towards the safety culture in the organization and 5 items of a more instrumental nature	10 items	87 Health quality referents (benchmarks) (HQR) 54 professionals working in smaller centers (with between 4 and 7 professionals) 100 professionals from the largest centers (more than 7 professionals)	Patient Continuity Adequacy Technical Competence Satisfaction Accessibility Clinical safety Equity	24 items
**Opinion Patients**			**Experience**		
**Stakeholder**	**Dimensions**	**Items**	**Stakeholder**	**Dimensions**	**Items**
Primary health centers: patients undergoing medical supervision of their disability (CC-common contingency-), patients treated for accidents at work (PC-professional contingency-), patients in rehabilitation (RHB) and patients in the emergency department (EMG)—(3.000 patients aprox.) Hospital centers: patients discharged, with admission to the hospitals. (1.500 patients approx.)	Patient Continuity Technical Competence Satisfaction Accessibility	Professional contingency: 27 items Common contingency: 25 items Rehabilitation: 23 items Emergency: 19 items	3 focus groups were conducted on patients in the mutual society hospital centers in Barcelona, Madrid and Seville	Patient Continuity Technical Competence Satisfaction Accessibility	Meetings in focus group

**Table 3 ijerph-16-03856-t003:** Overall satisfaction of patients in a mutual society (0–10 points). Period 2014–2017.

Services	2014	2015	2016	2017
Emergencies	8.5	8.3	8.4	9.0
Professional Contingency	8.7	8.7	8.5	8.7
Common Contingency	8.4	8.2	8.5	8.9
Rehabilitation	8.9	8.8	8.8	8.8
Hospitalization	9.0	9.0	9.0	9.1

**Table 4 ijerph-16-03856-t004:** Selection of Health Quality Plan Indicators.

Indicator
Result of opinion surveys on the overall evaluation of patients treated in hospital centers
Result of the opinion surveys of patients treated in specific Professional Contingency centers
Number of complaints and claims due to improper approach of personnel (Professional Contingency)
Number of complaints and claims due to improper approach of personnel (Common Contingency)
Hours of training in Patient Safety in healthcare professionals
Percentage of professionals trained in Health Quality
Number of health providers with quality agreements
Global assessment score of the culture survey
Percentage of excellent scores ≥9 in the global assessment of the culture survey

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
