# Peer review of "Experience of Good Practice in an Occupational Accident Mutual Insurance Society, Based on the Voice of Patients and Professionals"

_ijerph, 2019, doi:10.3390/ijerph16203856_

Round 1

Reviewer 1 Report

Thank you for the privilege of reviewing this detailed analysis of occupational safety practices, vulnerabilities, and opportunities for improvement.  This study contributes to existing research addressing country-specific cases (Park et al., 2019), alternative workers' compensation programs (Schofield, Ryan, & Dauner, 2019), demographic-specific situations (Argudo et al., 2019), and holistic perspectives (Tamers et al., 2019).

While focusing on "both patients and professionals" has value, an opportunity for improvement would be to expand the focus to include employer and/or organization perspectives.  As an example, Tamers et al. (2019) comment:

"To help organizations launch and sustain their own programs, the Office for TWH developed this workbook centered on five fundamental steps essential to the TWH approach. These five defining elements of TWH are guiding principles that provide practical direction for organizations seeking to develop workplace policies, programs, and practices that contribute to worker safety, health, and well-being:

-- Demonstrate leadership commitment;
-- Eliminate hazards and promote well-being;
-- Engage workers in program design and delivery;
-- Ensure confidentiality and privacy; and
-- Integrate systems effectively."

Employers and organizations are vital stakeholders in the occupational safety triumvirate:  worker, healthcare professional, and institutional leadership.  As such, their perspective should be addressed.  If not possible due to the study constraints, expanding the focus to include the employer and/or institutional component should be stipulated as a recommendation for future research.  

Thank you again, for the privilege of reviewing this intriguing study.

References

Argudo, E., Grehan, J., Leidy, L., Park, J. S. A., Patterson, M., Sanghavi, S., ... & Guerlain, S. (2019, April). Development and evaluation of an online ergonomics educational program for healthcare professionals. In 2019 Systems and Information Engineering Design Symposium (SIEDS) (pp. 1-6). IEEE.

Park, D. U., Choi, S., Lee, S., Koh, D. H., Kim, H. R., Lee, K. H., & Park, J. (2019). Occupational characteristics of semiconductor workers with cancer and rare diseases registered with a workers' compensation program in Korea. Safety and Health at Work, 347-354.

Schofield, K., Ryan, A. D., & Dauner, K. N. (2019). Comparing disability and return to work outcomes between alternative and traditional workers' compensation programs. American Journal of Industrial Medicine, 62(9), 755-765.

Tamers, S. L., Chosewood, L. C., Childress, A., Hudson, H., Nigam, J., & Chang, C. C. (2019). Total Worker Health® 2014–2018: The novel approach to worker safety, health, and well-being evolves. International journal of environmental research and public health, 16(3), 321.

Author Response

Dear reviewer:

Thank you very much for the comments received, for the references provided and for the employers' perspective which is not covered in the paper.

Thanks to their recommendations, we can include the professional perspective of the patient and the third client, which is the person or organization that assumes the economic expense but does not enjoy it.

Reviewer 2 Report

As you describe the data gathering process you may lose the reader a little bit in perhaps too much methodologic detail. Consider ways that you can depict more of this with another figure or table. table 2 numbers are drawn from Likert scale 0-10? you may need to indicate that on the side of the table.

Figure 1 needs to be broken up into smaller figures; it is really hard to navigate. 

Your section on the lessons learned needs to reflect more of the unique/novel parameters of the mutual society that may or may not be reproducible in other international contexts. Also, acknowledge more limitations of the methodology used. I understand from earlier in the paper that patient satisfaction surveys have not been employed for leading change to the same extent as some of the other measures. perhaps final sentences can integrate more details on what you describe in lines 347-350 rather than the platitude in 359. 

You need a citation for statistics noted in lines 36-37. 

Author Response

Dear reviewer, thank you very much for your comments that will help clarify this manuscript.

Here are the changes we have made that appear in red in the manuscript

As you describe the data gathering process you may lose the reader a little bit in perhaps too much methodologic detail. Consider ways that you can depict more of this with another figure or table.

We have tried to clarify the dimensions of safety culture so that they are not confusing (line 172-173); (line 222-227).

In addition to following your recommendations, we have made a table (table 2) summarizing each instrument, the items it has, who it is aimed at, the dimensions or areas it explores, with the aim of clarifying the methodology that has been used in each of the studies. We have included table 2, so the numbering of the rest of the tables change

table 2 numbers are drawn from Likert scale 0-10? you may need to indicate that on the side of the table.

We have included it (line 271)

Figure 1 needs to be broken up into smaller figures; it is really hard to navigate. 

Thank you for this suggestion, we have tried to make a much simpler figure for readers to understand.

Your section on the lessons learned needs to reflect more of the unique/novel parameters of the mutual society that may or may not be reproducible in other international contexts. Also, acknowledge more limitations of the methodology used. I understand from earlier in the paper that patient satisfaction surveys have not been employed for leading change to the same extent as some of the other measures. perhaps final sentences can integrate more details on what you describe in lines 347-350 rather than the platitude in 359. 

We have included our main limitation and that is that all are studies based on perception, but ultimately it is the goal to implement actions and see how they are perceived (line 363-365).

We have eliminated the line 359 and we have incorporated as end the lines 347-350

You need a citation for statistics noted in lines 36-37. Thanks, it´s the same citation than reference 2

Thanks for your review